# Selected Organ and Endocrine Complications According to BMI and the Metabolic Category of Obesity: A Single Endocrine Center Study

**DOI:** 10.3390/nu14061307

**Published:** 2022-03-20

**Authors:** Ewa Malwina Milewska, Ewelina Szczepanek-Parulska, Martyna Marciniak, Aleksandra Krygier, Agnieszka Dobrowolska, Marek Ruchala

**Affiliations:** 1Department of Endocrinology, Metabolism and Internal Diseases, Poznan University of Medical Sciences, 61-701 Poznan, Poland; ewamalwina.milewska@gmail.com (E.M.M.); ola-hernik@wp.pl (A.K.); mruchala@ump.edu.pl (M.R.); 2Department of Gastroenterology, Dietetics and Internal Diseases, Poznan University of Medical Sciences, 61-701 Poznan, Poland; marmarciniak@ump.edu.pl (M.M.); agdob@ump.edu.pl (A.D.)

**Keywords:** obesity, metabolically healthy obesity, metabolic syndrome, pre-metabolic syndrome, NAFLD, hypothyroidism

## Abstract

Obesity is a chronic and complex disease associated with metabolic, organ and endocrine complications. In the study, we analyzed a group of 105 patients suffering from obesity without any other previously recognized serious disorders who had been referred to a single endocrine center. The study aimed to assess the prevalence of selected organ and endocrine complications by subdividing the group, firstly according to body mass index (BMI) and secondly with regard to metabolic syndrome (MetS), pre-MetS and the metabolically healthy obesity (MHO) category. We have observed that in our groups, the prevalence of hyperlipidemia, hypertension, asthma, obstructive sleep apnea (OSA) depended on BMI category, whereas the incidence of hyperlipidemia, hypertension, OSA, hypothyroidism, non-alcoholic fatty liver disease, prediabetes, and type 2 diabetes was related to the metabolic category. We concluded that the distribution of particular organ and endocrine complications change significantly with increased BMI and with the shift from MHO to pre-MetS and MetS. Thus, to determine the risk of organ and endocrine complications more effectively, BMI and metabolic status should be assessed during the examination of patients with obesity.

## 1. Introduction

Obesity is a complex and chronic disease with no tendency to spontaneous regression, leading to numerous health complications. The WHO defines obesity as an excessive accumulation of fat tissue (men > 25%, women > 35%), with a body mass index (BMI) with a cut-off value of ≥30 kg/m^2^ [1]. Although this simplistic attempt to define and measure obesity is still used in the current guidelines, it has frequently been criticized [2]. In 2016, the AACE/ACE proposed a more comprehensive, complication-centric approach [3]. It distinguishes 4 grades of obesity according to BMI (≥25 kg/m^2^; ≥30 kg/m^2^) and the presence of weight-related complications caused by excess fat. In fact, excessive adiposity leads to obesity-related diseases, such as prediabetes, type 2 diabetes (DM2), dyslipidemia, hypertension, cardiovascular disease (CVD), non-alcoholic fatty liver disease/non-alcoholic steatohepatitis (NAFLD/NASH), polycystic ovary syndrome (PCOS), female infertility, male hypogonadism, obstructive sleep apnea (OSA), asthma and reactive airway disease, osteoarthritis, urinary stress incontinence, gastroesophageal reflux disease, and depression. Moreover, it is well established that the life expectancy of individuals suffering from obesity is reduced due to the aforementioned complications and the increased cancer risk [4,5].

The evaluation of patients with obesity should include the clinical assessment of possible weight-related complications, as well as anthropometric measurements, specifically BMI and waist circumference (WC). Interestingly, the measurement of WC in patients with a BMI < 35 kg/m^2^ provides extra information regarding metabolic risk and accumulation of visceral fat tissue [6,7]. Complementary methods, such as body bioimpedance, may also be crucial and allow for a precise and detailed assessment of the body composition; nevertheless, their availability in clinical practice is limited. According to the guidelines, laboratory tests should include fasting glucose, lipid profile, glucose tolerance, and liver function tests. Additionally, in case of clinical suspicion of hypercortisolism, male hypogonadism, fertility disturbances or hypopituitarism, a more detailed hormonal evaluation may be performed, which includes cortisol level measurement, 1 mg overnight dexamethasone suppression test, total testosterone, sex hormone-binding globulin (SHBG), follicle-stimulating hormone (FSH), luteinizing hormone (LH), androstenedione, estradiol, 17-OH-progesterone, prolactin, and insulin-like growth factor 1 (IGF-1) [2]. Furthermore, alterations in thyroid function may trigger the development of obesity in a predisposed group of patients, although the need for thyroid function assessment varies according to different guidelines. In fact, the European Guidelines for Obesity Management in Adults state that laboratory examinations should include thyroid function for every obese patient [2], while the AACE/ACE Guidelines for the Medical Care of Patients with Obesity suggest biochemical testing for hypothyroidism should be performed only in case of clinical suspicion [3].

Predispositions to certain complications may be related not only to BMI but also to body fat distribution. Many studies have confirmed that the accumulation of visceral adipose tissue, in particular, may lead to metabolic complications and the development of metabolic syndrome (MetS). MetS is comprised of a combination of factors that increase the risk of CVD and DM2. It is also characterized by chronic low-grade inflammation and a prothrombotic state, leading to endothelial dysfunction and atherosclerosis. The criteria have evolved over time. The most recent criteria regarding Mets were proposed in 2009 by consensus made by the International Diabetes Federation and the American Heart Association/National Heart, Lung, and Blood Institute. They include the measurement of WC, blood pressure (BP), triglycerides (TG), high-density lipoprotein cholesterol (HDL) and fasting glucose (FG) [8]. Three of the five abnormalities would qualify a person for MetS. Interestingly, although the cut-off for BP, TG, HDL and FG are fixed, those for WC are ethnic-specific. In fact, WC cut-off values for the Caucasian population are >84 cm and >90 cm for women and men, respectively [8]. However, neither insulin resistance nor NAFLD has been included in the current criteria of MetS, despite being significant risk factors for the development of DM2 and CVD.

Since neither BMI nor excessive fat mass is indispensable for the definition of MetS, individuals with normal BMI may be diagnosed with MetS. Similarly, some patients with obesity may not present any components of MetS. Thus, this subpopulation of obese patients without elevated CVD risk is referred to as metabolically healthy obese (MHO). The term MHO was first used by J. Vague in 1950 in his observational study [9], and since then, many heterogeneous definitions of MHO have emerged. Most recently, in 2018, a standardized version of the definition of MHO was proposed, according to which BMI ≥ 30 kg/m^2^ and none of the cardiometabolic risk factors are present, such as TG ≤ 150 mg/dL), HDL serum concentrations > 40 mg/dL (in men) or >50 mg/dL in women, systolic blood pressure (SBP) ≤ 130 mmHg, diastolic blood pressure (DBP) ≤ 85 mmHg, and FG ≤ 100 mg/dL. Furthermore, it is vital to note that MHO individuals should not receive antihypertensive, glucose-lowering or hypolipemic treatment [10]. Remarkably, all the metabolic criteria and cut-off values are exactly the same as in the current definition of MetS (excluding the criteria for WC, which has been replaced by BMI).

Although the prevalence of MHO differs depending on gender, region, and age, the BioSHaRE-EU Healthy Obese Project data estimate it at 12% of all cohorts [11]. This subphenotype of obesity is characterized by lower liver fat context, lower visceral adiposity, high leg fat mass, normal inflammatory markers, and higher cardiorespiratory fitness and activity [12,13,14]. Moreover, sleep disturbances are much less frequent among MHO [15]. Nevertheless, numerous studies indicate that MHO should rather be seen as a transitional state; in fact, in long-term observational studies, individuals classified as MHO developed metabolic alterations, shifting into the category of metabolically unhealthy obese (MUO) [16,17], and were at a higher risk of mortality due to cardiometabolic disease in comparison to the healthy, lean population [18]. Due to the fact that the transition between MHO and MetS is continuous, it is possible to distinguish the intermediate group of individuals who are not MHO but do not yet fulfill the criteria for MetS. These individuals are referred to as the pre-metabolic syndrome group (pre-MetS). This subgroup seems heterogeneous, not well-described (scarce publications, mainly regarding specific subpopulations, such as PCOS), with the potential of a two-directional transition (either to MetS or to MHO). Our study aimed to assess the prevalence of selected organ and endocrine diseases according to BMI and to MetS/pre-Mets/MHO category, with particular attention focused on the pre-MetS group.

## 2. Materials and Methods

This observational, monocentric study was conducted among patients with obesity in the Department of Endocrinology, Metabolism and Internal Medicine, Poznan University of Medical Sciences, Poznan, Poland, between September 2020 and December 2021.

The results of the anthropometric measurements, laboratory findings and prevalence of the selected obesity-related and endocrine diseases were analyzed in the group of consecutive patients suffering from obesity, willing to undergo treatment, who were referred to the department to exclude a secondary endocrine cause of obesity and to introduce therapy.

The studied group consisted of 105 patients (90 females and 15 males) aged 18–62 years, with the BMI ranging between 30–56 kg/m^2^. In order to increase group homogeneity, we included only treatment-naive patients (without previous pharmacological or surgical attempts at weight reduction). Additionally, to simplify the inclusion criteria, we defined obesity as BMI ≥ 30 kg/m^2^. In contrast, the exclusion criteria included patients with BMI < 30 kg/m^2^; aged < 18 years, lactation or pregnancy, active neoplastic disease, severe pre-existing systemic (infectious, autoimmunologic, endocrinological, psychiatric) diseases, systemic steroid therapy up to one month prior to the initial evaluation, Cushing’s disease, pre-existing diabetes, and previous pharmacological or surgical obesity treatment. All patients were in generally good health or with well-controlled diseases, such as hypothyroidism, hypertension or dyslipidemia.

Following an overnight fast, all patients had their BP measured. The other tests included a complete blood count, lipid profile, liver function tests, gonadotropins, concentrations of estradiol and androgens, SHBG, prolactin, assessment of thyroid function, vitamin D, cortisol and adrenocorticotropic hormone. Insulin and glucose measurements were performed fasting and after 2 h following the oral administration of 75 g of glucose, which is known as an oral glucose tolerance test (OGTT). Depression screening was conducted according to Beck Depression Inventory. All patients were consulted by a qualified dietician. The assessment of the selected obesity-related and endocrine diseases was made according to a detailed interview and evaluation of laboratory findings. Additionally, the Homeostatic Model Assessment for Insulin Resistance (HOMA-IR) was calculated [19].

In the studied patient group, we analyzed the following anthropometric measurements: BMI and WC; the following laboratory measurements: total cholesterol (TC), HDL, TG, low-density lipoprotein cholesterol (LDL), aspartate aminotransferase (AST), alanine aminotransferase (ALT), gamma-glutamyl transferase (GGTP), and the HOMA-IR; and the following selected metabolic and endocrine complications: hypertension, prediabetes, DM2, NAFLD, asthma, OSA, depression, male hypogonadism, PCOS, infertility, and hypothyroidism. BMI was calculated on the basis of the WHO criteria. WC was measured in the upright position, placing the elastic tape above the iliac crests. PCOS diagnosis was confirmed by a gynecologist-endocrinologist according to the ESHRE guidelines [20]. Male hypogonadism was diagnosed on the basis of testosterone measurement, FSH, LH and SHBG, as well as an interview with regard to mood disorders and erectile and other sexual dysfunctions. NAFLD was diagnosed on the basis of elevated AST, ALT, GGTP and/or steatosis confirmed by ultrasound, exclusion of alcohol abuse, or hepatitis. Depression was evaluated according to the Polish adaptation of the Beck Depression Inventory with a cut-off value of >11, indicating possible depression and the need for further psychological consultation [21]. The assessment of infertility, OSA, and asthma was based on the clinical interview.

We divided the subjects into 3 groups according to BMI (grade I obesity: BMI 30–34.9 kg/m^2^; grade II obesity: BMI 35–39.9 kg/m^2^; grade III obesity: BMI ≥ 40 kg/m^2^) and according to metabolic complications (pre-MetS, MetS and MHO). MetS was diagnosed in accordance with the 2009 consensus established by the International Diabetes Federation and the American Heart Association/National Heart, Lung, and Blood Institute [11], when patients met at least 3 of the following criteria: WC  ≥  80 cm in females and ≥94 cm in males; TG concentration  ≥ 150 mg/dL or treatment for elevated TG; HDL concentration  < 40 mg/dL in males or <50 mg/dL in females, or treatment for low HDL; SBP ≥ 130 mmHg or DBP  ≥ 85 mmHg, or previously diagnosed high BP; and FG  ≥ 100 mg/dL, or treatment for elevated glucose levels. The Pre-MetS group met at least one MetS criteria (excluding WC), although they failed to fulfill all the MetS criteria. All subjects who did not meet any MetS criteria (excluding WC) and did not receive antihypertensive, glucose-lowering or hypolipemic treatment were considered MHO [10].

The study’s protocol was in accordance with the 1964 Helsinki Declaration and its subsequent amendments or comparable ethical standards and was approved by the Poznan University of Medical Sciences Bioethics Committee (approval number: 279/21). Informed consent was obtained from all participants of the study.

### Statistical Analysis

Statistical analysis of the acquired data was performed using STATISTICA software-version 13 (StatSoft, Tulsa, OK, USA) and TIBCO Software Inc.(Palo Alto, CA, USA) (2017). The data distribution was evaluated by the Shapiro–Wilk test. Due to the lack of a normal distribution of all parameters, nonparametric tests were applied. The comparison between the groups according to metabolic status and obesity grade was performed using the Mann–Whitney U test and ANOVA Kruskal–Wallis test. Chi-squared and R Spearman tests were applied to evaluate the correlation between the variables. The data are presented as a median and 25–75% interquartile range [IQR], or mean values and SD. The level of statistical significance was set at *p*  < 0.05.

## 3. Results

The analyzed group consisted of 105 participants, including 90 women and 15 men. The median age was 34 (26~44) years. The highest percentage of participants presented with grade I obesity (43%), followed by grade II obesity (35%) and grade III obesity (22%), according to BMI. The prevalence of MetS was 45%. The prevalence of pre-MetS was 38%, and the prevalence of MHO was 17%, according to the metabolic group. A statistically significant association was found between obesity grade and the metabolic category. The subjects with grade I obesity were more frequently included in the MHO and pre-MetS categories, whereas the participants with grade II and III obesity were more often classified as MetS (Figure 1A,B).

We observed statistically significant differences between the groups divided according to BMI in WC, HOMA-IR, ALT, and AST (with the lowest values found in grade I obesity and the highest in grade III obesity). No statistically significant differences were found in the serum levels of TC, TG, HDL, LDL, VLDL, or fasting glucose, or after 2 h in OGTT and GGTP (Table 1). When comparing grade I obesity vs. grade II and grade III obesity groups, the only parameters presenting a statistically significant difference were WC (*p* < 0.001) and the HOMA-IR (*p* = 0.001). A graphical presentation of the selected parameters is depicted in Figure 2.

The prevalence of hypertension, OSA, asthma and hyperlipidemia depended on the BMI category. Hypertension, OSA and asthma were statistically less frequent in subjects with grade I obesity in comparison to grade III obesity (Table 2). When we compared grade I obesity vs. grade II and grade III obesity, we additionally observed that hyperlipidemia was significantly less frequent in grade I obesity (Table 3).

When the patients were subdivided according to a metabolic group, we found statistically significant differences for WC, TC, HDL, LDL, VLDL, ALT, AST, GGTP, fasting glucose and after 2 h in OGTT and the HOMA-IR between the groups (Table 4). The MetS individuals presented significantly increased values of WC, TG, LDL, VLDL, GGTP and HOMA-IR in comparison to MHO and pre-MetS, a significantly higher level of AST, ALT in comparison to MHO and a significantly decreased HDL as compared to MHO. The pre-MetS presented significantly higher TG, VLDL and HOMA-IR in comparison to the MHO category. The MHO individuals showed decreased WC, TC, LDL, ALT, AST, GGTP and HOMA-IR in comparison to MetS and pre-MetS. A graphical illustration of the most statistically significantly different parameters is depicted in Figure 3 and Figure 4.

Following a subdivision of the studied group according to the metabolic category (pre-MetS/MetS/MHO), we noticed significant statistical differences between the groups in terms of the prevalence of hyperlipidemia, hypertension, OSA, prediabetes, DM2 de novo, NAFLD, and hypothyroidism. Although the MHO subjects showed the lowest incidence of the aforementioned complications, they still presented such obesity-related diseases as PCOS, depression, infertility, male hypogonadism, and asthma (Table 5). Figure 5 demonstrates complications presenting the most statistically significant differences between particular metabolic categories.

Subsequently, the complications which were found to differ significantly between the subgroups divided according to BMI and the metabolic category were correlated to all the performed anthropometric and laboratory measurements. The presented statistically significant associations were as follows. In the entire studied group, we found that patients with hyperlipidemia were characterized with a higher WC than individuals with normal lipid levels (106.91 ± 10.55 vs. 101.55 ± 11.3, *p* = 0.02). Patients with NAFLD showed both higher values of BMI (38.14 ± 5.14 vs. 35.1 ± 4.38, *p* = 0.002) and WC (110.05 ± 10.65 vs. 101.27 ± 10, *p* < 0.001). Subjects with prediabetes presented both higher BMI (37.33 ± 4.67 vs. 35.66 ± 5.21, *p* = 0.03) and WC (109.26 ± 10.04 vs. 101.82 ± 10.85, *p* = 0.001). Patients suffering from hypertension had both higher BMI (37.64 ± 4.06 vs. 35.75 ± 5.35, *p* = 0.01) and WC (108.48 ± 10.16 vs. 103.21 ± 11.18, *p* = 0.02). Moreover, participants with hyperlipidemia were characterized with higher HOMA-IR values (6.88 ± 5.15 vs. 4.86 ± 3.27 *p* = 0.03). Subjects with NAFLD presented increased HOMA-IR values (8.7 ± 5.48 vs. 4.31 ± 2.77, *p* < 0.001).

The parameters of NAFLD (AST, ALT, GGTP) correlated with atherogenic dyslipidemia and augmented LDL levels. GGTP levels positively correlated with TG (r = 0.39, *p* < 0.001) and LDL (r = 0.24; *p* = 0.02), while negatively correlating with HDL (r = −0.24, *p* = 0.01). Similarly, ALT correlated positively with TG (r = 0.22, *p* = 0.03) and LDL (r = 0.26, *p* = 0.03), although it correlated negatively with HDL (r = −0.21; *p* = 0.004). AST correlated positively with LDL (r = 0.24, *p* = 0.02).

## 4. Discussion

In our study, we identified three subgroups of the patients depending on the presence of metabolic complications: MHO, MetS and pre-MetS. The total prevalence of MHO was 17%, with a distinct difference according to the BMI category, which is in accordance with the BioSHaRE-EU Healthy Obese Project, which estimates it at 12% [11]. This group was characterized by lower HOMA-IR scores, a better lipid profile and better liver enzymes. In terms of obesity-related diseases, the group showed the lowest prevalence of hypothyroidism, OSA, NAFLD, and dyslipidemia, although obesity-related complications were still present, including depression, infertility, PCOS and male hypogonadism. The subgroup identified as MetS, apart from demonstrating a higher incidence of complications which, per se, constituted the components of metabolic syndrome (hypertension, prediabetes/diabetes, hyperlipidemia, high WC), showed the poorest outcome with respect to the highest BMI, HOMA-IR, as well as the highest prevalence of OSA, NAFLD and hypothyroidism. Interestingly, we also managed to distinguish the subgroup defined as pre-MetS. Thus far, the literature concerning the pre-MetS subgroup has been rather scarce, and the previous studies have focused on specific subgroups, such as PCOS, and have mainly been conducted in Asia or South America [22,23,24,25]. The inherent difficulty has been related to the heterogeneous nature of the group, as it has been characterized with a significantly higher HOMA-IR and TG in comparison to MHO, and with significantly lower WC and LDL and GGTP when compared to MetS. This, in turn, accounts for the reason why we were able to find more significant differences when analyzing the group according to the metabolic category rather than the BMI. Moreover, it is in line with the studies indicating that BMI and the metabolic state, to a larger extent, contribute to the development of such complications as prediabetes, DM2, hyperlipidemia and, in consequence, CVD [26,27]. Additionally, predisposition to certain complications is related to body fat distribution. Although data considering the predictive factors of the metabolic shift from MHO to MetS are limited and contradictory, the common risk factors in these studies are insulin resistance (IR)/higher HOMA-IR and higher visceral adiposity/higher WC [26,27,28]. In fact, the main underlying cause of MetS is excessive fat tissue, particularly abdominal obesity, which is strongly associated with impaired insulin resistance and fatty liver disease [6,7].

NAFLD is one of the most common chronic liver diseases, which is increasing in incidence and constitutes one of the leading causes of liver transplantation. It is very strongly associated with hepatic and peripheral insulin resistance, which impairs the utilization of free fatty acids, leading to their excessive accumulation in hepatocytes and steatosis. In the studied group, the prevalence of NAFLD was the highest in patients with grade III obesity, although the differences between the groups divided according to BMI did not reach statistical significance. Moreover, the subjects with NAFLD presented significantly higher WC, HOMA-IR and BMI. In addition, we found that GGTP and ALT were associated with atherogenic dyslipidemia (higher TG and lower HDL) and higher LDL, which significantly contributes to increased CVD risk. It is crucial to note that our results are in concordance with the conclusions of other studies indicating that NAFLD might be regarded as an independent risk factor for the development of such serious metabolic complications as DM2 and CVD [29,30,31,32,33]. Despite the fact that NAFLD is not included in the MetS criteria, the assessment of liver function contributes to a better metabolic assessment and may help to classify the patient according to the cardiometabolic risk.

In this context, it is also essential to assess thyroid function in patients with obesity since thyroid dysfunction is associated with excessive adiposity, musculoskeletal deterioration and MetS [34]. Both clinical and subclinical hypothyroidism may contribute to a decreased metabolic rate, abnormal body fat distribution and weight gain [35,36,37]. Isolated elevated thyroid-stimulating hormone (TSH) is a common finding in patients suffering from obesity. TSH stimulates the release of leptin, which promotes the conversion of thyroxine (T4) to triiodothyronine (T3) [38]. In turn, decreased free T4 levels are associated with unfavorable changes in the lipid profile, as well as muscular and liver accumulation of fatty acids, IR and, in consequence, the development of MetS [39,40]. Furthermore, hyperinsulinemia itself has the potential for goiter and nodule formation [41]. In our study, we did not report a statistically significant relationship between the prevalence of hypothyroidism and the BMI category. However, it was strongly associated with the metabolic category and the lowest prevalence among the MHO population. The possible explanation may be linked to IR; the differences in the HOMA-IR were more pronounced between the metabolic categories in comparison to the subgroups divided according to BMI. Moreover, thyroid dysfunction exerts a more significant impact on metabolic function than body weight itself. Recent studies have also linked hypothyroidism with NAFLD in the subjects suffering from morbid obesity and patients with DM2 [42,43]. However, when analyzing the subjects with hypothyroidism, we did not find BMI, WC or HOMA-IR to be statistically significantly higher in the hypothyroid group.

Unfavorable changes in lipid profile and progressive insulin resistance negatively affect hepatic function and the development of hypertension. In our study, the incidence of hypertension positively correlated with the BMI category. However, the most distinct differences in the prevalence were observed when the patients were subdivided according to the metabolic category. These results are in accordance with other studies, which reported a correlation between BMI and hypertension [44,45,46,47]. Additionally, hypertension is strictly related to the markers of visceral adiposity, such as WC and insulin resistance [46,47], and these parameters were statistically significantly higher in the MetS group. However, in our study, we found a significant association between BMI and WC, but not with HOMA-IR. The pathomechanism of hypertension is complex, and other factors, including genetics, may play a pivotal role. In fact, the risks of hypertension and poorly controlled hypertension significantly increase in patients suffering from OSA. Although the group with the diagnosed OSA was relatively small (10 subjects from our group), its prevalence significantly correlated with BMI and the metabolic category, with its highest incidence in grade III obesity and the MetS group. Interestingly, 60% of patients with OSA suffered from hypertension, 90% of whom presented with NAFLD, and 90% developed MetS, which may point to the severity of the disease and the multidirectional relation between the complications.

To the best of our knowledge, this is the first publication concerning the assessment of MHO in the central European population that focuses on organ and endocrine complications among patients with obesity who previously perceived themselves as healthy. Furthermore, it is also the first publication aiming at the characterization of a pre-MetS subpopulation of patients suffering from obesity.

It is also necessary to mention certain limitations of the study. Firstly, it was monocentric and included a relatively small cohort size. However, we carefully selected patients who did not suffer from any serious diseases and were treatment-naive. Secondly, the study would benefit from including more patients, which would possibly result in a larger male population, MHO subgroup and subjects with grade III obesity, who were underrepresented.

We paid particular attention to the anthropometric measurements, such as BMI and WC, as well as basic biochemical measurements (liver enzymes, TSH, HOMA-IR, lipid profile), which are easily accessible in everyday clinical practice. However, there are a number of other key directions for further clinical and basic science research. More detailed studies, including genetic, immunological and biomolecular factors, could contribute to elucidating the pathogenesis of obesity and the associated complications. In the future, observations should involve the evaluation of body composition, diet and physical activity patterns, which may significantly contribute to the metabolic category, as well as the presence of certain complications.

## 5. Conclusions

The interrelationship between endocrine and metabolic complications of obesity is complex and multidirectional. The prevalence of particular organ and endocrine complications changes significantly not only with the increase of BMI but primarily with the shift from MHO to pre-MetS and MetS. Therefore, not only BMI but also metabolic status should be assessed during the examination of patients with obesity in order to determine the risk of organ and endocrine complications more effectively, as well as to individualize the treatment. Furthermore, it is critical to identify the pre-MetS group among patients with obesity where the applied intervention may potentially prevent the development of full MetS.

## Figures and Tables

**Figure 1 nutrients-14-01307-f001:**
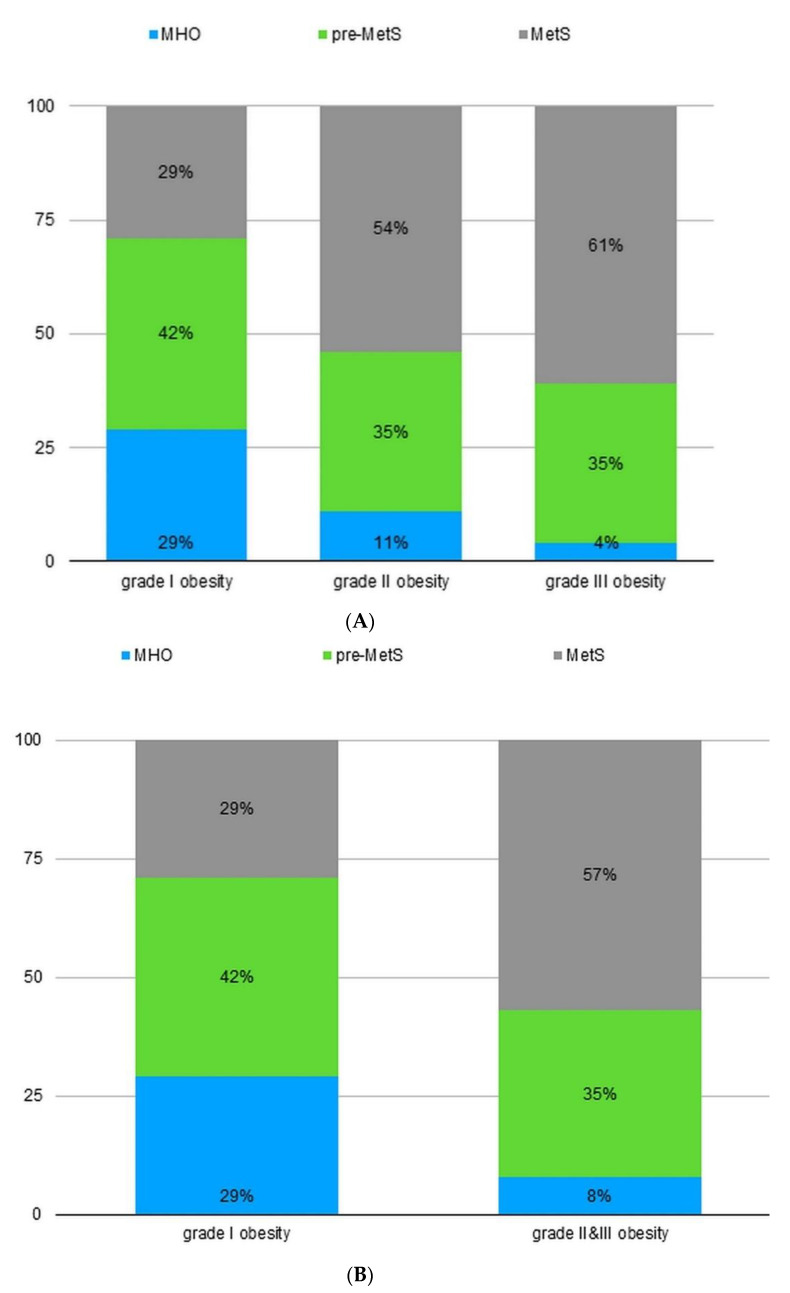
(**A**) The proportion of patients from three metabolic categories (MHO, pre-MetS, MetS) according to the obesity grade (three groups). MHO: metabolically healthy obesity; pre-MetS: pre-metabolic syndrome; MetS: metabolic syndrome. (**B**) The proportion of patients from three metabolic categories (MHO, pre-MetS, MetS) according to the obesity grade (two groups). MHO: metabolically healthy obesity; pre-MetS: pre-metabolic syndrome; MetS: metabolic syndrome.

**Figure 2 nutrients-14-01307-f002:**
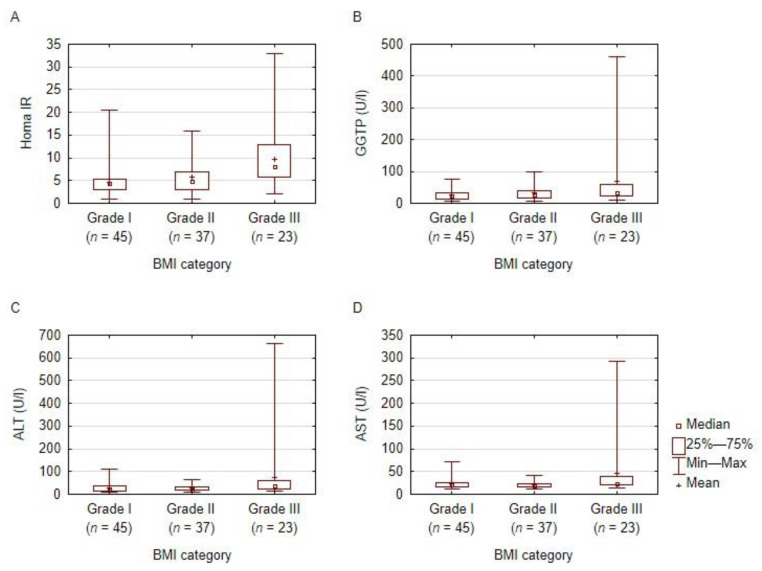
HOMA-IR and liver enzymes according to BMI. A graphical presentation of the distribution of the HOMA-IR (box **A**), GGTP (box **B**), AST (box **C**), and ALT (box **D**) among the subjects, according to BMI category. BMI: body mass index; ALT: alanine transaminase; AST: aspartate transaminase; GGTP: gamma-glutamyl transferase; HOMA-IR: Homeostatic Model Assessment for Insulin Resistance; grade I obesity: BMI 30–34.9 kg/m^2^; grade II obesity: BMI 35–39.9 kg/m^2^; grade III obesity: BMI ≥ 40 kg/m^2^.

**Figure 3 nutrients-14-01307-f003:**
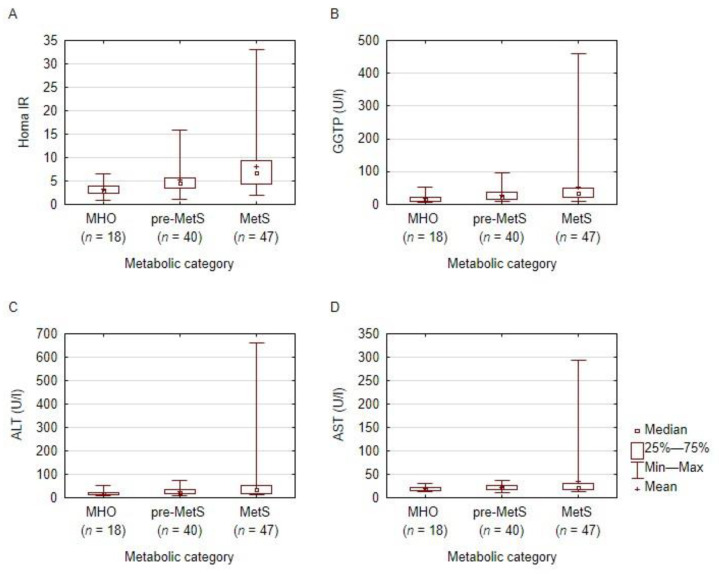
HOMA-IR and liver enzymes according to the metabolic category. A graphical presentation of HOMA-IR (box **A**), GGTP (box **B**), AST (box **C**), ALT (box **D**) distribution among the subjects, according to the metabolic category. ALT: alanine transaminase; AST: aspartate transaminase; GGTP: gamma-glutamyl transferase; HOMA-IR: Homeostatic Model Assessment for Insulin Resistance. MHO: metabolically healthy obesity; pre-MetS: pre-metabolic syndrome; MetS: metabolic syndrome.

**Figure 4 nutrients-14-01307-f004:**
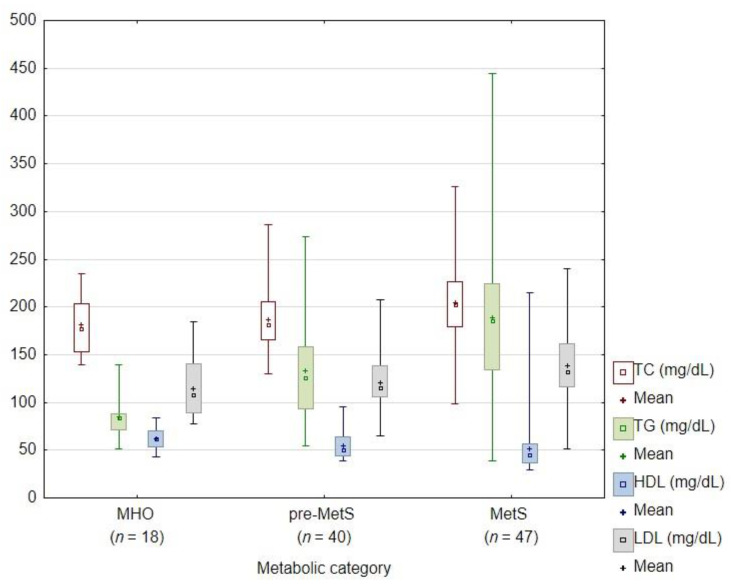
Cholesterol fractions according to the metabolic category. A graphical presentation of the distribution of cholesterol fractions among the subjects according to the metabolic category. TC: total cholesterol; TG: triglycerides; HDL: high-density lipoprotein cholesterol; LDL: low-density lipoprotein cholesterol; MHO: metabolically healthy obesity; pre-MetS: pre-metabolic syndrome; MetS: metabolic syndrome.

**Figure 5 nutrients-14-01307-f005:**
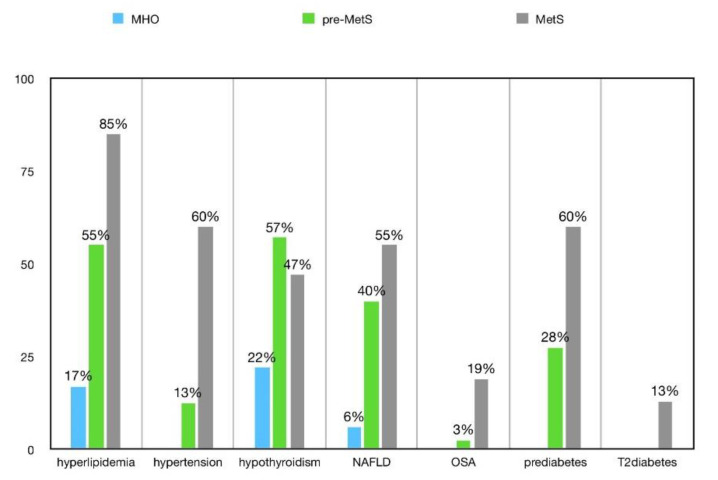
A graphical presentation of the statistically important selected organ and endocrine disorders according to the metabolic category. MHO (*n* = 18), pre-MetS (*n* = 40), MetS (*n* = 47). MHO: metabolically healthy obesity; pre-MetS: pre-metabolic syndrome; MetS: metabolic syndrome; NAFLD: non-alcoholic fatty liver disease; OSA: obstructive sleep apnea.

**Table 1 nutrients-14-01307-t001:** Anthropometric measurements and laboratory findings according to the BMI category.

Parameter	Grade I Obesity *n* = 45 Me (Q1–Q3)	Grade II Obesity *n* =37 Me (Q1–Q3)	Grade III Obesity *n* =23 Me (Q1–Q3)	H	*p*
Age (years)	34 (26–44)	37 (28–46)	28 (25–41)	3.85	0.15
WC (cm)	98 (94–104)	104 (99–110)	121 (113–124)	41.76	**<0.001**
TC (mg/dL)	186 (164.5–206.5)	189 (171–218)	190 (172–224)	0.54	0.76
TG (mg/dL)	113.5 (83.5–166)	146 (114–189)	160 (103–209)	3.47	0.18
HDL (mg/dL)	52.5 (43–67.5)	51 (44–63)	45 (41–55)	3.32	0.19
LDL (mg/dL)	115.5 (99–143.5)	126.5 (105–145.5)	126 (116–159)	2.11	0.35
VLDL (mg/dL)	22 (16.6–33)	29.2 (22.8–37.8)	32 (20.6–41.8)	4.11	0.13
ALT (U/I)	21 (15–35)	21 (17–32)	36 (25–59)	11.90	**0.003**
AST (U/I)	21 (17–26)	19 (16–23)	24 (20–40)	9.04	**0.01**
GGTP (U/I)	22 (15–34)	27 (16–39)	33 (23–61)	5.75	0.06
G-0	96 (91–99)	96 (93–104)	98 (91–113)	1.64	0.44
G-120	109 (96–130)	110 (93–126)	130 (85–153)	1.79	0.41
HOMA-IR	4.4 (2.9–5.3)	4.85 (3–6.9)	8.1 (5.7–12.8)	19.60	**<0.001**

H-ANOVA Kruskal–Wallis test, a comparison between 3 groups; values are expressed as a median (Me) and an interquartile range (Q1–Q3). A *p*-value in bold type marks a significant difference (*p* < 0.05). BMI: body mass index; WC: waist circumference; TC: total cholesterol; TG: triglycerides; HDL: high-density lipoprotein cholesterol; LDL: low-density lipoprotein cholesterol; VLDL: very-low-density lipoprotein cholesterol; ALT: alanine transaminase; AST: aspartate transaminase; GGTP: gamma-glutamyl transferase; G-0: fasting glucose; G-120: glucose after two hours in oral glucose tolerance test; HOMA-IR: Homeostatic Model Assessment for Insulin Resistance.

**Table 2 nutrients-14-01307-t002:** BMI category and prevalence of the selected obesity-related or endocrine disorders (3 groups).

Disorder	Grade I Obesity *n* = 45	Grade II Obesity *n* = 37	Grade III Obesity *n* = 23	X^2^	*p*
Hyperlipidemia	51.11%	67.57%	73.91%	4.16	0.12
Hypothyroidism	42.22%	43.24%	60.87%	2.4	0.30
Depression	42.22%	43.24%	39.13%	0.10	0.95
NAFLD	31.11%	40.54%	60.87%	5.56	0.06
Prediabetes	35.56%	37.84%	56.52%	2.96	0.23
Type 2 diabetes de novo	6.67%	5.41%	4.35%	0.17	0.92
Hypertension	15.56%	45.95%	39.13%	9.98	**0.01**
PCOS	42.5%	34.29%	46.67%	0.86	0.65
Infertility	5%	11.43%	13.33%	1.45	0.48
Male hypogonadism	80%	100%	87.5%	0.75	0.69
OSA	2.22%	10.81%	21.74%	7.02	**0.03**
Asthma	0%	2.7%	13.04%	6.98	**0.03**

X^2^: Chi-squared test for *p*-value, a comparison between three groups. A *p*-value in bold type marks a significant difference (*p* < 0.05). BMI: body mass index; NAFLD: non-alcoholic fatty liver disease; PCOS: polycystic ovarian syndrome; OSA: obstructive sleep apnea.

**Table 3 nutrients-14-01307-t003:** BMI category and prevalence of the selected obesity-related or endocrine disorders (two groups).

Disorder	Grade I Obesity *n* = 45	Grade II & III Obesity *n* = 60	X^2^	*p*
Hyperlipidemia	51.11%	70%	3.89	**0.049**
Hypothyroidism	42.22%	50%	0.63	0.43
Depression	42.22%	41.67%	0.00	0.95
NAFLD	31.11%	48.33%	3.19	0.07
Prediabetes	35.56%	45%	0.95	0.33
Type 2 diabetes de novo	6.67%	5%	0.13	0.72
Hypertension	15.56%	43.33%	9.71	**0.002**
PCOS	42.5%	38%	0.19	0.67
Infertility	5%	12%	1.42	0.23
Male hypogonadism	80%	90%	0.27	0.60
OSA	2.22%	15%	5.73	**0.02**
Asthma	0%	6.67%	4.60	**0.03**

X^2^: Chi-squared test for *p*-value, a comparison between three groups. A *p*-value in bold type marks a significant difference (*p* < 0.05). BMI: body mass index; NAFLD: non-alcoholic fatty liver disease; PCOS: polycystic ovarian syndrome; OSA: obstructive sleep apnea.

**Table 4 nutrients-14-01307-t004:** Anthropometric measurements and laboratory findings according to the metabolic category.

Parameter	pre-MetS *n* = 40 Me (Q1–Q3)	MetS *n* = 47 Me (Q1–Q3)	MHO *n* = 18 Me (Q1–Q3)	H	*p*
Age (years)	30 (24.5–39.5)	38 (27–45)	40 (30–46)	2.81	0.25
BMI	35.5 (32–39)	37 (34–40)	32.5 (31–35)	15.82	**<0.001**
WC (cm)	99.5 (95–110)	108 (100–120)	96 (91–101)	22.47	**<0.001**
TC (mg/dL)	181 (165.5–205)	202.5 (179–226)	177 (153–203)	8.11	**0.02**
TG (mg/dL)	125.5 (93.5–158)	186 (134–224)	83.5 (71–88)	33.27	**<0.001**
HDL (mg/dL)	50.5 (44–64)	45.5 (37–57)	62 (53–70)	14.32	**0.001**
LDL (mg/dL)	115.5 (106–138.5)	132 (116–161)	108 (89–141)	9.75	**0.01**
VLDL (mg/dL)	25.1 (18.7–31.6)	36.6 (26.6–44.8)	16.7 (14.2–17.6)	30.58	**<0.001**
ALT (U/I)	23.5 (17–33.5)	33 (19–52)	18 (14–22)	13.12	**0.001**
AST (U/I)	21.5 (18–26.5)	21 (17–31)	18 (15–22)	6.96	**0.03**
GGTP (U/I)	22 (15–36)	33 (23.5–51.5)	16 (11–23)	19.57	**<0.001**
G-0 (mg/dL)	95 (92–100)	99 (94–108)	94 (88–98)	9.54	**0.01**
G-120 (mg/dL)	109 (88–119)	126 (107–165)	102.5 (93–113)	14.74	**<0.001**
HOMA-IR	4.5 (3.6–5.75)	6.75 (4.4–9.4)	2.95 (2.4–3.95)	21.98	**<0.001**

H-ANOVA Kruskal–Wallis test, a comparison between 3 groups as a median (Me) and an interquartile range (Q1–Q3). A *p*-value in bold type marks a significant difference (*p* < 0.05). BMI: body mass index; WC: waist circumference; TC: total cholesterol; TG: triglycerides; HDL: high-density lipoprotein cholesterol; LDL: low-density lipoprotein cholesterol; ALT: alanine transaminase; AST: aspartate transaminase; GGTP: gamma-glutamyl transferase; HOMA IR: Homeostatic Model Assessment for Insulin Resistance.

**Table 5 nutrients-14-01307-t005:** The metabolic category and prevalence of the selected obesity-related or endocrine disorders.

Disorder	pre-MetS *n* = 40	MetS *n* = 47	MHO *n* = 18	X^2^	*p*
Hyperlipidemia	55%	85.11%	16.67%	28.72	**<0.001**
Hypothyroidism	57.5%	46.81%	22.22%	6.51	**0.04**
Depression	37.5%	44.68%	44.44%	0.52	0.77
NAFLD	40%	55.32%	5.56%	15.92	**<0.001**
Prediabetes	37.5%	59.57%	0%	25.76	**<0.001**
Type 2 diabetes de novo	0%	12.77%	0%	10.10	**0.01**
Hypertension	12.5%	59.57%	0%	37.16	**<0.001**
PCOS	48.72%	28.57%	43.75%	3.29	0.19
Infertility	5.13%	14.29%	6.25%	2.03	0.36
Male hypogonadism	100%	83.33%	100%	0.97	0.62
OSA	2.5%	19.15%	0%	10.78	**0.005**
Asthma	2.5%	4.26%	5.56%	0.37	0.83

X^2^: Chi-squared test for *p*-value, a comparison between three groups. A *p*-value in bold type marks a significant difference (*p* < 0.05). MHO: metabolically healthy obese; pre-MetS: pre-metabolic syndrome; MetS: metabolic syndrome; NAFLD: non-alcoholic fatty liver disease; PCOS: polycystic ovarian syndrome; OSA: obstructive sleep apnea.

## Data Availability

The data presented in this study are available on request from the corresponding author.

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
