# Peer review of "Selected Organ and Endocrine Complications According to BMI and the Metabolic Category of Obesity: A Single Endocrine Center Study"

_nutrients, 2022, doi:10.3390/nu14061307_

Round 1

Reviewer 1 Report

The manuscript submitted by Milewska and co-authors offers an interesting preliminary study on role of body mass and obesity on co-morbidities and associated disorders. Unfortunately major limitations for the proposed study stand on the monocentric a small cohort of patients (approx. 100 patients collected within 16mos). 

The amount of analysis and hemathological parameters currently offered are quite limited. Biomolecular analysis, genetic investigation on predisposition, and immune cytological analysis are highly recommended for such study where autoimmune disorders have been correlated with patients conditions.

Results representation (Fig.1) may largely benefit from revision and editing: why Y axis are up to 1.25?

Table 1-5 requires SD/SEM and a graphical representation, maybe through whiskey-plot or similar, including n values and eventually dots, would be important to determine variability and distribution

Importantly, the conclusions stated in the title do not completely fit with the results nor can be supported by the limited measurements and results collected. The authors should avoid over-statement and too ambitious conclusion based on limited data and patients

Minor details:

line 39: CVD acronym needs to be elucidated.  

Author Response

Poznan, 26 Feb 2022

Dear Editor, 

Thank you for your e-mail and the reviewers comments to our submission Manuscript ID: nutrients-1598651. We are grateful to the reviewers for all valuable suggestions. According to the reviewers’ comments we have revised point by point our manuscript. The responses to the reviewers’ notes are listed below. All the changes made in the manuscript are marked in color.

Yours sincerely,

on my behalf and co-authors,

Ewelina Szczepanek-Parulska

Reviewer 1

The manuscript submitted by Milewska and co-authors offers an interesting preliminary study on role of body mass and obesity on co-morbidities and associated disorders. Unfortunately major limitations for the proposed study stand on the monocentric a small cohort of patients (approx. 100 patients collected within 16mos). 

Thank you for your comment and appreciation of our work. We are aware that our results would be more valuable if larger cohort of patients were analyzed. However, we aimed to have well selected group of patients - obese but without any previously known serious disease. In our endocrinological department we deal with a very wide range of patients. Due to the specificity of our center (that serves as reference center for several regions of the country) most of the patients have concomitant disorders, that are listed as exclusion criteria. Of course in the future we will try to confirm our findings on a larger group of subjects. In the discussion section of our revised paper we have underlined more this important limitation of our study. We have also modified the title and conclusions to sound more appropriate and less confident, having in mind this limitation.

The amount of analysis and hemathological parameters currently offered are quite limited. Biomolecular analysis, genetic investigation on predisposition, and immune cytological analysis are highly recommended for such study where autoimmune disorders have been correlated with patients conditions.

Thank you for your important comment that puts our study in a broader perspective. Our patients have had done many biochemical analysis of all parameters that are performed in the clinical practice during evaluation of metabolic status of obese patients and that are currently recommended to exclude endocrine causes of weight gain. We have used those parameters to establish diagnosis of the organ and endocrine complications analyzed in our study. However, the concept of our study was to concentrate on relatively simple and basic measurements (anthropometric: WC, BMI and laboratory tests routinely performed) that are easily obtained in everyday practice, but allow to classify patients according to their metabolic and endocrine status necessary for adjustment of the therapy. Thus, the results obtained from our observations can be easily adopted to clinical routine. Undoubtedly, there are many different factors potentially playing important role in pathogenesis of obesity and related complications. We have included your suggestion in the in the limitations sections of the manuscript. In the ongoing observation we are planning to perform more detailed investigation of the group focusing i.e. on endogenous peptides that are potentially linked to obesity and metabolic complications.

Results representation (Fig.1) may largely benefit from revision and editing: why Y axis are up to 1.25?

Thank you for your comment. The graphs have been modified and the presentation of the results was revised.

Table 1-5 requires SD/SEM and a graphical representation, maybe through whiskey-plot or similar, including n values and eventually dots, would be important to determine variability and distribution

The suggested changes have been made. We have added n values and SEM and interquartile range where required, moreover we have included new figures: Figure 2 A,B,C,D and 3 A,B,C,D that correspond to the results presented in Table 1 and Table 4. We have added Figure 4 that corresponds to the results included in the Table 5. Thank you for your comment - the graphical presentation of the results is much improved after changes.

Importantly, the conclusions stated in the title do not completely fit with the results nor can be supported by the limited measurements and results collected.

Thank you very much for the comment. We have changed the title of the paper for the one that more appropriately reflects the content of our manuscript. The new title proposed sounds as follows:

The prevalence of selected organ and endocrine complications according to BMI and metabolic category of obesity - a single endocrine center study

The authors should avoid over-statement and too ambitious conclusion based on limited data and patients

The conclusions were modified accordingly. :

We concluded that the distribution of particular organ and endocrine complications differs significantly not only with the increase of BMI, but especially with the shift from MHO via pre-MetS to MetS category. Thus, the assessment of not only BMI but also metabolic category should be performed during the routine assessment of obese patients for better stratification of the risk of associated organ and endocrine complications.

Minor details:

line 39: CVD acronym needs to be elucidated.  

Thank you for your comment. The necessary explanation has been done in the text.

Reviewer 2

  1. the Abstract need to revise /edit to more concisely, too many repeat words/terms.

Thank you for your comment. We took into account your suggestions and have modified the abstract to be more concise and stylistically appropriate.

  1. The Figure/table need to add a brief legend.

Thank you very much for the comment. We have added legends to tables and figures and explained all abbreviations.

Reviewer 2 Report

  1. the Abstract need to revise /edit to more concisely, too many repeat words/terms.
  2. The Figure/table need to add  a brief Legend.

Author Response

Poznan, 26 Feb 2022

Dear Editor, Professor Jia Yi Huin

Thank you for your e-mail and the reviewers comments to our submission Manuscript ID: nutrients-1598651. We are grateful to the reviewers for all valuable suggestions. According to the reviewers’ comments we have revised point by point our manuscript. The responses to the reviewers’ notes are listed below. All the changes made in the manuscript are marked in color.

Yours sincerely,

on my behalf and co-authors,

Ewelina Szczepanek-Parulska

Reviewer 1

The manuscript submitted by Milewska and co-authors offers an interesting preliminary study on role of body mass and obesity on co-morbidities and associated disorders. Unfortunately major limitations for the proposed study stand on the monocentric a small cohort of patients (approx. 100 patients collected within 16mos). 

Thank you for your comment and appreciation of our work. We are aware that our results would be more valuable if larger cohort of patients were analyzed. However, we aimed to have well selected group of patients - obese but without any previously known serious disease. In our endocrinological department we deal with a very wide range of patients. Due to the specificity of our center (that serves as reference center for several regions of the country) most of the patients have concomitant disorders, that are listed as exclusion criteria. Of course in the future we will try to confirm our findings on a larger group of subjects. In the discussion section of our revised paper we have underlined more this important limitation of our study. We have also modified the title and conclusions to sound more appropriate and less confident, having in mind this limitation.

The amount of analysis and hemathological parameters currently offered are quite limited. Biomolecular analysis, genetic investigation on predisposition, and immune cytological analysis are highly recommended for such study where autoimmune disorders have been correlated with patients conditions.

Thank you for your important comment that puts our study in a broader perspective. Our patients have had done many biochemical analysis of all parameters that are performed in the clinical practice during evaluation of metabolic status of obese patients and that are currently recommended to exclude endocrine causes of weight gain. We have used those parameters to establish diagnosis of the organ and endocrine complications analyzed in our study. However, the concept of our study was to concentrate on relatively simple and basic measurements (anthropometric: WC, BMI and laboratory tests routinely performed) that are easily obtained in everyday practice, but allow to classify patients according to their metabolic and endocrine status necessary for adjustment of the therapy. Thus, the results obtained from our observations can be easily adopted to clinical routine. Undoubtedly, there are many different factors potentially playing important role in pathogenesis of obesity and related complications. We have included your suggestion in the in the limitations sections of the manuscript. In the ongoing observation we are planning to perform more detailed investigation of the group focusing i.e. on endogenous peptides that are potentially linked to obesity and metabolic complications.

Results representation (Fig.1) may largely benefit from revision and editing: why Y axis are up to 1.25?

Thank you for your comment. The graphs have been modified and the presentation of the results was revised.

Table 1-5 requires SD/SEM and a graphical representation, maybe through whiskey-plot or similar, including n values and eventually dots, would be important to determine variability and distribution

The suggested changes have been made. We have added n values and SEM and interquartile range where required, moreover we have included new figures: Figure 2 A,B,C,D and 3 A,B,C,D that correspond to the results presented in Table 1 and Table 4. We have added Figure 4 that corresponds to the results included in the Table 5. Thank you for your comment - the graphical presentation of the results is much improved after changes.

Importantly, the conclusions stated in the title do not completely fit with the results nor can be supported by the limited measurements and results collected.

Thank you very much for the comment. We have changed the title of the paper for the one that more appropriately reflects the content of our manuscript. The new title proposed sounds as follows:

The prevalence of selected organ and endocrine complications according to BMI and metabolic category of obesity - a single endocrine center study

The authors should avoid over-statement and too ambitious conclusion based on limited data and patients

The conclusions were modified accordingly. :

We concluded that the distribution of particular organ and endocrine complications differs significantly not only with the increase of BMI, but especially with the shift from MHO via pre-MetS to MetS category. Thus, the assessment of not only BMI but also metabolic category should be performed during the routine assessment of obese patients for better stratification of the risk of associated organ and endocrine complications.

Minor details:

line 39: CVD acronym needs to be elucidated.  

Thank you for your comment. The necessary explanation has been done in the text.

Reviewer 2

  1. the Abstract need to revise /edit to more concisely, too many repeat words/terms.

Thank you for your comment. We took into account your suggestions and have modified the abstract to be more concise and stylistically appropriate.

  1. The Figure/table need to add a brief legend.

Thank you very much for the comment. We have added legends to tables and figures and explained all abbreviations.

Round 2

Reviewer 1 Report

We are quite satisfied with the modification and clarification the authors offered in R1 version. However, A final revision and editing for English language is recommended

Check Ref nr. 14, 44 and 47. Pages are missing

We still consider hematological analysis for blood lipids (total cholesterol, high-density lipoprotein, low-density lipoprotein, very low-density lipoprotein and triglycerides) and blood sugar relevant and pertinent to the proposed study

Author Response

Poznan, 16th March 2022

Dear Editor, Prof. Jia Yi Huin

Thank you for the reviewer’s comments to our submission Manuscript ID: Nutrients-1598651. We followed the suggestions and introduced some changes in the manuscript - they are marked in yellow color. The responses to the reviewer’s notes are listed below.

Yours sincerely,

on my behalf and co-authors,

Ewelina Szczepanek-Parulska

Comments and Suggestions for Authors

We are quite satisfied with the modification and clarification the authors offered in R1 version. However, A final revision and editing for English language is recommended

Check Ref nr. 14, 44 and 47. Pages are missing

We still consider hematological analysis for blood lipids (total cholesterol, high-density lipoprotein, low-density lipoprotein, very low-density lipoprotein and triglycerides) and blood sugar relevant and pertinent to the proposed study

Thank you for your valuable comments and appreciation of our work. We have taken into account your suggestions and included the missing pages in the References nr. 14, 44, 47. We have added the new Figure representing the differences in the distribution of cholesterol fractions (total cholesterol, HDL-cholesterol, LDL-cholesterol and triglycerides) according to metabolic category (Figure 4). In our laboratory we do not obtain the results of VLDL levels but as it is derived from triglycerides level we have added this value to the text and to the Table 1 and 4, using Friedewald’s formula. To keep the graphical presentation clear we did not include it into the Figure 4. We decided not to add the figure representing the cholesterol fractions distribution according to BMI category as there are no statistically important differences. These data have been already included in the tables but now it is much more clear. We added to the Table 1 and Table 4 the claimed values of glucose measurement: fasting and after 2 hours in oral glucose tolerance test.

Thank you for your suggestion of linguistic correction that we have followed; now we believe the text is much improved.